# Amine Detection Using Organic Field Effect Transistor Gas Sensors

**DOI:** 10.3390/s21010013

**Published:** 2020-12-22

**Authors:** Panagiotis Mougkogiannis, Michael Turner, Krishna Persaud

**Affiliations:** 1Department of Chemical Engineering and Analytical Science, The University of Manchester, Manchester M13 9PL, UK; panagiotis.mougkogiannis@manchester.ac.uk; 2Department of Chemistry, The University of Manchester, Manchester M13 9PL, UK; michael.turner@manchester.ac.uk

**Keywords:** organic field effect transistor, gas sensor, DPP-T-TT, ammonia, alkylamines, empirical model

## Abstract

Low power gas sensors with high sensitivity and selectivity are desired for many practical applications. Devices based on organic field effect transistors are promising because they can be fabricated at modest cost and are low power devices. Organic field effect transistors fabricated in bottom-gate bottom-contact configuration using the organic semiconductor [2,5-(2-octyldodecyl)-3,6-diketopyrrolopyrrole-alt-5,5-(2,5-di(thien-2-yl)thieno] [3,2-b]thiophene) (DPP-T-TT) were systematically investigated to determine the response characteristics to a series of alkylamines and ammonia. The highest sensitivity was to dibutylamine with a limit of detection of 0.025 ppb, followed by n-butylamine, 0.056 ppb, and ammonia, 2.17 ppb. A model was constructed based on the Antoine equation that successfully allows the empirical prediction of the sensitivity and selectivity of the gas sensor to various analytes including amines and alcohols based on the Antoine C parameter and the heat of the vaporization of the analyte.

## 1. Introduction

Conjugated and conducting polymers have a long history of over 150 years as reviewed by Rasmussen [1]. Organic semiconducting polymers (OSCs) are described in terms of the energy bands that originate from the bonding and anti-bonding energy levels associated with the σ-bonds between adjacent carbon atoms (formed from the sp^2^ wavefunctions) and the orthogonal π-bonds that originate from the p_z_ wavefunctions. The σ-bonds are important in holding the structure together, but the π-bonds are the origin of the properties that characterize conjugated polymers as semiconductors. The discovery of highly conducting polyacetylene in 1977 with the first true evidence of conductive polymers exhibiting conductivity comparable to metals led to the recognition of Heeger, MacDiarmid and Shirakawa with the Nobel prize in 2000 [2]. The range of organic semiconductor materials available has expanded tremendously and these materials are now ubiquitous in consumer electronics [3] with organic field effect transistors (OFETs) and organic light emitting diodes (OLEDs) being a fundamental electronic building blocks of organic electronic circuits. 

There has been much interest in utilizing organic semiconductors as gas sensors, due to perceived limitations and disadvantages of the currently available technology. The most common commercially available semiconductor gas sensors are based on porous metallic oxide materials, typically such as ZnO, SnO_2_, WO_3_ and Fe_2_O_3_ [4]. However, they have limitations due to poor selectivity and high operating temperature despite recent advances in micromachined devices and smart gas sensing approaches, as reviewed by Feng et al. [5]. The most selective sensors commonly used are electrochemical gas sensors that are relatively limited in the range of gases that they can detect [6]. This is despite the many types of gas sensors developed in research laboratories that include resonator and mechanical devices, optical devices and others as reviewed by Hunter et al. [7], which have had limited commercial success. The driving forces for the development of OSC-based gas sensors are due to the great flexibility of design possible by the selection of OSCs appropriate for target analytes. Gas sensors based on organic field effect transistors (OFETs) are promising for industrial use because they are amenable to solution processable manufacturing techniques, allowing mass production at modest prices without the added cost associated with clean room techniques [8]. 

This paper focuses on OFET sensors targeted towards ammonia and amine sensing. The compounds are of great interest due to their relevance in environmental monitoring, food safety and healthcare applications. Ammonia sensing has received a great research attention, due to the key role of ammonia in severe respiratory diseases. Ammonia is commonly used in many industries, including petrochemical, pulp and paper, and fertilizers. Ammonia detection also finds application in environmental monitoring in the food industry. For example, ammonia concentration should be kept below 20 ppm in chicken farms to prevent respiratory diseases and secondary infections [9]. Gas sensors that are capable of detecting ammonia at ppb to low ppm levels are not readily available commercially. Metal oxide ammonia sensors such as the TGS826 from Figaro, Japan, reach a lower limit of 30 ppm [10], electrochemical sensors such as that from Winsen Electronics, Japan [11], are usable in the range between 10 and 100 ppm with a zero drift between 3 and 10 ppm. 

Organic semiconductor gas sensors normally use the π-conjugated materials as the active sensing layer that functions both as a transducer as well as a receptor. The device configuration could either be a two-terminal resistor [12] or a three-terminal field effect transistor (FET) [13] with the electrical properties being affected owing to the interaction between the analyte and the active semiconductor layer. Device parameters such as conductivity, mobility or threshold voltage vary with the concentration of the analytes adsorbed. For an organic semiconductor to function as a chemical sensor, it is necessary to cause some perturbation of the electronic properties upon the interaction with a molecule that may serve as a secondary dopant, so if a charge transfer complex is formed between the analyte molecule and the OSC, by either donating or accepting a fractional charge, then a signal is transduced. Janata and Josowicz [14] pointed out that the exposure to a gas or vapour introduces changes, in analogy to inorganic semiconductors, in the occupancy level at the valence band edge and the conduction band edge, resulting in the variation of the energy at the Fermi level, E_F_. They observe that a new equilibrium state in the semiconductor is established by this secondary doping that is governed by the solubility properties of the analyte in the semiconductor. The volatile analyte–semiconductor equilibrium is governed by the charge transfer equilibrium of this system and obeys Henry’s law:(1)$$K_{G}=\;\frac{{\left[e\right]}^{2\partial }}{\propto P_{G}}$$
where K_G_ is the equilibrium constant, P_G_ is the partial pressure of the analyte, α is the solubility of the analyte G in the solid phase, and δ is the fractional charge of electron transferred from the analyte to the polymer. 

Here, we focus on organic field effect transistors. Such devices as illustrated in Figure 1 comprise π-conjugated organic molecules or polymers working as active channel materials, organic or inorganic insulators serving as dielectric layers with metals or carbon materials acting as source, drain, gate electrodes. The active OSC layer is located in the channel between the source and drain electrodes and is isolated from the gate electrode by a dielectric layer. OFETs are characterized by their output and transfer characteristics. A plot of source–drain current (I_DS_) versus source–drain voltage (V_DS_) at different but constant source–gate voltages (V_GS_) is used to describe the output, while a plot of I_DS_ versus V_GS_ at constant V_DS_ is used to describe the transfer characteristics. The threshold voltage, V_th_, is defined as the minimum V_GS_ required to turn on the transistor. In the linear region (V_DS_ < V_GS_ − V_th_), the drain current I_DS_ depends on V_GS_ and V_DS_ (Equation (2)):(2)$$I_{\mathrm{DS}}=\frac{{\mathrm{WC}}_{i}\mu }{L}\left(V_{\mathrm{GS}}-V_{\mathrm{th}}-\frac{V_{\mathrm{DS}}}{2}\right)V_{\mathrm{GS}}$$
and in the saturated regime (V_DS_ > V_GS_ − V_th_) it depends on Equation (3):(3)$$I_{\mathrm{DS}}=\frac{{\mathrm{WC}}_{i}\mu }{2L}{\left(V_{\mathrm{GS}}-V_{\mathrm{th}}\right)}^{2}$$
where I_DS_ is the source–drain current, W is the width and L is the length of the channel, respectively, μ is the field effect mobility, C_i_ is the capacitance per unit area, V_GS_ is the gate voltage and V_th_ is the threshold voltage. This makes an OFET essentially a multiparametric sensing system. When V_GS_ = 0 or less than V_th_, the OFET is considered to be in the “off” state, and any drain current at this point is essentially the intrinsic current of the OSC. At this stage, the analyte will essentially permeate through the OSC and interact with its bulk. Any resulting change in the intrinsic conductivity is recorded as the measure of sensitivity of that particular analyte. This mechanism is basically the same mechanism as in the chemiresistive sensors. During the “on” regime of the OFET, the I_DS_ flows through the 2-dimensional conduction channel and the interaction between the analyte and the active OSC not only influences the I_DS_ but also affects the other parameters like the threshold voltage and field effect mobility. Thus, “on” state and “off” mechanisms are entirely different, and both can supplement the sensing information in a complementary fashion, a clear advantage over the traditional chemiresistors. However, as Janata and Josowicz point out, the introduction of the analyte vapour can result in the net change of the carriers in the OSC, leading to a change in work function and the conductivity of the layer and the height of the Schottky barrier at the contact can also be modulated [14]. If the analyte is electroactive, it can change the charge transfer resistance at the drain contact, by acting as redox species at that junction. Since several mechanisms can operate simultaneously, it is difficult to understand the nature of the sensor response.

Progress in OFET gas sensor development has been extensively reviewed [8,13,15,16]. OSCs used are predominantly p-type and range from metal phthalocyanines, perylene derivatives to thiophene derivatives including conjugated polymers such as poly 3-hexylthiophenes and diketopyrrolopyrrole (DPP) derivatives. Exposure to oxidising gases normally increases the conductivity while reducing gases will have the converse effect. There are fewer n-type materials documented since they are often unstable for operation in air. OFET gas sensors developed to date unfortunately possess inherent drawbacks such as the lack of air stability due to the susceptibility of the sensing functionalities to interaction with moisture and oxygen under ambient conditions, lack of selectivity and a baseline drift problem, which limits the range of applications. However, this situation has changed dramatically due to the application of new types of OSC as described below.

Until recently, the majority of OFET devices investigated as gas sensors operated at rather high voltages e.g., an ammonia OFET reported by Rajeev et al. based on regioregular poly (3-hexylthiophene) (rr-P3HT) operated at a gate voltage of −35 V and V_DS_ of −40 to −45 V [17]_._ Similarly OFET ammonia sensors based on extremely promising diketopyrroles reported by Yang et al. [18] operated at V_GS_ and V_DS_ of −60 V. It has only been recently that devices that operate at low gate-source voltages with power consumption of the order of microwatts have been developed giving great advantages over conventional gas sensors based on metal oxides that require heating to high temperatures for normal operation [19]. 

This paper builds on the previous work of our group regarding OFET devices for ammonia sensing where Tate et al. [20] and Rahmanudin et al. [21] reported low voltage OFETs operating with V_GS_ and V_DS_ of −3 V. These were fabricated based on the organic semiconducting polymer (OSC) poly [2,5-(2-octyldodecyl)-3,6-diketopyrrolopyrrole-alt-5,5-(2,5-di(thien-2-yl)thieno] [3,2-b]thiophene) (DPPT-TT) deposited in conjunction with a chemically robust high k gate dielectric material of large areal capacitance, which were capable of detecting and measuring ammonia with high sensitivity. DPP-based materials are emerging as powerful materials that are applicable to a range of organic–semiconductor applications due to excellent planarity and better electron-withdrawing ability over other p-type materials and are reviewed by Liu et al. [22]. For ammonia sensing, DPP-T-TT displays high sensitivity and selectivity which is likely to be based on the interaction between ammonia and pyrrole, first observed by Kanazawa and Diaz [23] and later attributed to acid-base interactions between ammonia and the polymer [24]. The bottom-gate bottom-contact (BGBC) configuration shown in Figure 1 allows the OSC to be exposed to the analyte vapour while all other components of the OFET are protected. 

As alluded to earlier, the nature of the molecular interactions that confer sensitivity and selectivity to different analytes for OFET gas sensors is poorly understood. The OSC interacts with adsorbed analyte vapour via physical and chemical interactions that perturb the output of the OFET to transduce a measurable electrical response. Analyte interaction is based on the geometrical characteristics of the devices and the characteristics of the OSC and can be grouped in terms of hydrogen bonding, charge transfer, hydrophobic, hydrophilic and dipole–dipole interactions, van der Waals attraction, etc. In general, the OFET gas sensor performance can be expressed in terms of the charge carrier mobility μ, threshold voltage V_th_ and the on–off current ratio (I_ON_/I_OFF_). For ammonia, without the aid of chemical reactions, the molecules influence the charge transport by adsorbing on the polymers through the Coulomb interaction, which reduces the conductance in p-type polymers; and accumulating at the polymer/dielectrics interface, which changes the charge distribution at the interface and leads to threshold voltage (V_th_) shifts [25]. The focus of this study is to investigate the effect of the analyte–polymer interactions through systematic characterization of low voltage operation OFETs based on DPP-T-TT as OSC, fabricated in bottom-gate bottom-contact (BGBC) configuration, for detecting ammonia and alkyl-amine vapours. We show that the OSC interactions with analytes can be predicted on the basis of physico-chemical characteristics of the analyte molecules and that an empirical model can be constructed on the basis of the experimental data.

## 2. Materials and Methods

### 2.1. Chemicals and Reagents

The chemicals used, (poly(vinylidenefluoride-trifluoroethylene-chlorofluoroethylene)) (P(VDF-TrFE-CFE)), poly(methyl methacrylate) (PMMA), benzophenone (BP), dichlorobenzene (DCB) and pentafluoro-benzenethiol (PFBT) were purchased from Sigma Aldrich. *N*,*N*-dimethylformamide (DMF) and anisole were purchased from Merck. [2,5-(2-octyldodecyl)-3,6-diketopyrrolopyrrole-alt-5,5-(2,5-di(thien-2-yl)thieno] [3,2-b]thiophene) (DPPT-TT) was synthesized in house using a reaction scheme described previously [26]. All solutions were filtered through a 0.45 µm filter just before use.

### 2.2. Sensor Fabrication

Vacuum thermal evaporation and spin coating (Laurell WS-650HZ-23NPP) were employed to fabricate a fully solution processed low voltage operation organic field effect transistor (OFET). The devices were fabricated on polyethylene naphthalate (PEN) film. PEN thickness is 125 μm and was purchased from Teijin Film Solutions. The PEN was fully bonded to a glass surface (24 mm × 24 mm × 1 mm from Agar Scientific) with a cool off tape (Plafix Intelimer from Nitta). The process for fabricating an OFET array was previously described [20,27]. 

An Al gate electrode was deposited in a thermal evaporator system to produce a film of 50 nm thickness. A high k dielectric solution consisting of P(VDF-TrFE-CFE) was prepared by weight 5% (*w*/*w*) in DMF. This was deposited on the gate electrode by spin coating to achieve a film thickness of 180 nm and was followed by the deposition of a low k dielectric consisting of 2% PMMA in anisole to achieve a film thickness of 30 nm. Au source and drain electrodes were deposited in a thermal evaporator system by placing a shadow mask between the target material and the substrate. The thickness of the source (Au) and drain (Au) electrodes was about 50 nm. When the deposition was finished the sample was immersed in a 5 mM PFBT/EtOH solution for 2 min. This step (contact modification) helps the injection of charge carriers from contacts to OSC. The OSC solution was prepared as DPP-T-TT by weight 0.5% (*w*/*w*) in DCB. This was spin coated between the source and drain electrodes to give a film thickness of 50 nm. The channel dimensions were 2000 µm width × 60 µm length.

### 2.3. Morphology

The surface morphology of the OSC deposited on the OFET devices was characterized using a Bruker Multimode 8 Atomic Force Microscope.

### 2.4. Gas Sensing Measurements

Ammonia and amine vapours at ppb levels were generated using permeation tubes calibrated gravimetrically and an Owlstone Permeation Oven (Owlstone OVG-4). This was mixed with carrier gas (dry air or humidified air) using a series of mass flow controllers in an automated gas rig as described previously [28]. A humidifier (Owlstone OHG-4) provided moist air whenever it was needed to mix with the carrier stream. For the generation of concentrations at ppm levels, a standard calibration gas (1000 ppm ammonia in air) was used together with mass flow controllers to generate the desired concentrations.

The electrical response of the fabricated sensors toward different amine vapour concentrations in dry air and humid air was measured as the change in the current between the source and drain electrodes for a fixed gate-source voltage using an in-house designed microcontroller-based data acquisition system with a range of 0–2 µA and 10 pA resolution. 

Typically, the OFET sensor was exposed to pulses of ammonia at different concentration steps for periods of 5 min followed by recovery in clean air for 30 min. The change in current between drain and source electrodes (I_DS)_ was measured at each step. The sensor response is expressed as % change in I_DS_ which allows for comparison between devices that may have differing baseline I_DS:_(4)$$\mathrm{Response}=100\times \frac{I_{\max}-I_{\mathrm{baseline}}}{I_{\mathrm{baseline}}}$$
where I_max_ is the maximum change in current observed for a particular concentration of analyte after the given exposure time.

## 3. Results

Figure 1 illustrates the OSC, schematic of the BGBC device, fabricated OFET sensor array as well as the measurement circuit with a header covering the sensor array. The edge connectors on the devices Figure 1c allowed the insertion into a zero-insertion-force (ZIF) socket. As shown in Figure 1d, a header over the sensor array served as a chamber for introducing vapours to the sensor array in a controlled way. 

The inlet was connected to a gas rig comprising mass flow controllers for gas dilution that allowed for controlled concentrations of vapour to be presented to the OFET sensor array. The measurement of I_DS_ when devices were exposed to vapours was carried out via a custom designed electronic circuit with a microcontroller (Figure 1d) that allowed a multiplexed operation of the array of OFETS—applying an appropriate V_GS_ (−3 V in this case) measuring the current between the source and drain electrodes to a 10 pA resolution at appropriate V_DS_. This was connected via Universal Serial Bus (USB) to an external computer for real-time data acquisition and processing.

### 3.1. Morphology

Tapping mode AFM (Figure 2) revealed homogeneous nano-grain morphology with a low RMS roughness surface roughness R_q_ of 0.42 nm.

This is consistent with observations that sensing layers with smoother surfaces have higher field effect mobility and stability. To achieve high charge carrier mobilities requires minimal conformational disorder along the conjugated backbone [29]. The improved backbone co-planarity leads to the formation of high-mobility charge transport pathways in the polymer films, and high molecular weights promote high mobility [30]. Zhang et al. [31] have studied the molecular packing of DPP-based materials and conclude that there is extraordinary in-plane orientation of the polymer main chains, irrespective of whether the conjugated plane was edge-on or face-on. They suggest that this structural characteristic, together with the closely correlated local ring co-planarity, is responsible for the high mobilities found and that bulky side chains do not disrupt the organisation of the conjugated backbone of the polymer.

### 3.2. Electrical Characterisation

These OFET gas sensors were fabricated using solution processable techniques and operate at low voltages. OFETs based on DPPT-TT with Rq < 1 nm operated at V_DS_ = 3 V with minimal hysteresis, threshold voltage was −0.5 V, the on/off ratio of I_ON_/I_OFF_ ≅ 9.2 x 10^3^ and the value for the sub-threshold swing extracted from the transfer curve (Figure 3a) was 770 mV decade^−1^. The output curve (Figure 3b) displayed good operational characteristics from a V_GS_ of −1 to −3 V. The capacitance per unit area C_i_ was 50 nF cm^−2^ and at room temperature the field effect mobility (μ) was 2.5 ± 0.3. 10^−1^ cm^2^ (Vs)^−1^. The density of interfacial trap states N_it_ calculated was = 3.75. 10^12^ eV^−1^ cm^−2^. 

In organic electronics, the activation energy is defined as the energy difference between the transport level E_μ_ and the Fermi level E_F_ in the organic semiconductor. Hence, the field effect mobility in an organic device is determined by the activation energy. An Arrhenius plot (Figure 3c) indicated that conductivity in the OSC is thermally activated and follows an Arrhenius relationship. Here, an E_a_ of 82 ± 8 meV was calculated which is comparable to that calculated by Tanaka et al. [32] from field-induced electron spin resonance (ESR) spectroscopy measurements, of 68 meV. 

### 3.3. Response to Analyte Vapours

The response to ammonia and amines is manifested by a reversible decrease in I_DS_ when the OFET is operated in the saturation regime. The response kinetics, as measured by the change in source–drain current of the OFET, to ammonia and alkyl-amines are rather slow and similar to that recorded by other researchers for similar types of OSC, e.g., poly (3,3′′′-didodecylquaterthiophene) (PQT-12) and 2,2′-[(2,5-dihexadecyl-3,6-dioxo-2,3,5,6-tetrahydropyrrolo[3,4-c]pyrrole-1,4-diy-lidene)dithiene-5,2-diylidene] dimalononitrile (DPP-CN) [33]. As shown in Figure 4a, when exposed to a fixed concentration of ammonia (21 ppb) for increasing periods of time, followed by recovery in clean air, the change in I_DS_ is proportional to the length of exposure. Observing the rate of change of the response to a fixed concentration of ammonia over an extended period (Figure 4b, shows that this tends towards a steady state). This is attributed to a difference between the rate of association versus the rate of dissociation of the analyte from the OSC, the rate of dissociation being slower than the rate of association, as seen in Figure 4a where the time taken for recovery to baseline is much longer than the one response time—so that the sensor effectively accumulates the analyte. This is advantageous as the change in source–drain current could be controlled by the time of exposure to a fixed concentration of analyte gas or vapour. Limiting exposure to a fixed period (in this case 5 min), using a solenoid valve to switch analyte to the sensor array gave consistent reversible responses and the data in this paper are based on a 5 min exposure time for each analyte tested. Figure 4c shows the raw responses to decreasing concentrations of ammonia and Figure 4d a linear concentration–response relationship to low levels (ppb) of ammonia for a particular OFET device. However, the device has in fact a large dynamic range of up to several hundreds of ppm, and as shown in Figure 5a, the concentration–response relationship being nonlinear at the higher concentrations and best fitted by a parabola given by a second order polynomial. Variations in the OSC film thickness between different fabricated devices gave differences in sensitivity to the analyte vapour and a thicker film was less sensitive than a thinner film. Here, a nominal film thickness of ~50 nm was adopted.

Individual OFETs varied in baseline I_DS_ current after the manufactured devices typically had a range between 100 to 600 nA in clean air just after manufacture. These differences are due to variations in film thickness in dielectric and OSCs that can be made more consistent in the future by applying bulk manufacturing methods. These variations did not affect the responses seen to analytes as shown in Figure 5b where two transistors with differing I_DS_ base currents were exposed repeatedly and simultaneously to the same ammonia concentrations (57 ppm). For all OFET gas sensor devices, there is a gradual decrease in baseline I_DS_ over time and Figure 5c illustrates the change observed for one device over a period of 5 months. The device still responded to analyte vapours consistently over this time. The lifetime of such devices generally depends on how they are treated—exposure to solvents at saturated vapour concentrations will generally shorten the lifetime. The usability of such devices often depends on the resolution of the measurement system and successful measurements have been made with devices with baseline currents much lower than 100 nA.

The responses to a series of alkyl-amines, trimethylamine, triethylamine, n-propylamine, n-butylamine and dibutylamine over a range of concentrations were investigated. Figure 6a illustrates the raw responses for one OFET device to a series of exposures between 62 and 1052 ppb for triethylamine showing that the changes in I_DS_ were proportional to the concentration presented and that the current returned to a baseline after purging with clean air. Figure 6b shows the resulting concentration–response curve, the gradient being defined as “sensitivity”. Similar data were collected for all amines tested with replicated OFET devices and measured sensitivities and a calculated limit of detection (LOD) are given in Table 1. It is observed that the highest sensitivity among amines is to dibutylamine and the lowest is to triethylamine, while the sensitivity to alcohols was much lower.

### 3.4. Effect of Molecular Parameters of the Analyte to the Perceived Selectivity of the OSC

There are four distinctive forces of energy that influence the interaction of an analyte with the OSC: the dispersion forces such as van der Waals interactions, polar forces, hydrogen bonding, and ionic interaction. Hildebrand and Scott [34] argued that the solubility parameter δt is connected with the evaporation energy ΔΕ_V_ (Equation (5)). To better understand solvent properties, Hansen et al. (1967) [35] classified the Hildebrand solubility parameter into three distinct types including the dispersion forces (δD), polar forces (δP) and hydrogen bonding (δH) (Equations (6) and (7). Here, V_m_ is the molecular volume and CED is the cohesive energy:(5)$$\delta t=\sqrt{\frac{\Delta E_{v}}{V_{m}}}=\frac{\Delta H_{v}-\mathrm{RT}}{V_{m}}=\surd \left(\mathrm{CED}\right)$$
(6)$$E=E_{D}+E_{P}+E_{H}$$

Dividing Equation (5) by the molecular volume, the solubility parameter δt can be written as
(7)$${\delta t}^{2}={\delta D}^{2}+{\delta P}^{2}+{\delta H}^{2}$$

To attempt to correlate the relative sensitivity of the OSC for different analytes, the response curves to varying concentrations were measured for ammonia, a series of alkyl amines and two alcohols, producing similar data to those illustrated in Figure 4 and Figure 6. The measured sensitivities derived from the gradient of the concentration–response curves for three replicate OFETs are summarised in Table 1 (column 6) and Figure 7. They illustrate the high selectivity of DPP-T-TT OSCs to dibutylamine against the other compounds investigated, which could not previously have been predicted.

The molecular parameters consisting of the dispersion forces (δD), polar forces (δP) and hydrogen bonding (δH) are well documented. These were taken from Abbott et al. [36] and are shown in Table 1. These parameters were investigated to determine whether a quantitative structure–activity model (QSAR) could be applied to relate these findings to properties of the molecules tested. As shown in Figure 8a, where log(Sensitivity) is plotted against δD, there were no obvious correlations seen and this was also the case for δP and δH (data in Table 1).

Antoine (1888) [37] introduced an equation able to predict the vapour pressure of pure liquids (vaporization) and solids (sublimation) and this equation is still widely used today because of its accuracy. In this case it was observed that Antoine’s C constant (AntC) in degrees Celsius and the heat of vaporization at boiling point (ΔH_V_) correlated well with the observations of sensitivity shown in Figure 7. The Antoine equation describes the relation between the vapour pressure and the temperature of pure substances, as shown in Equation (8):(8)$$\mathrm{logP}=A-\frac{B}{T+C}$$
where P is the vapour pressure, T is the temperature, A, B and C are component-specific constants. Table 1 summarises AntC and ΔH_V_. A regression model y = A + Bx1 + Bx2, where x1 = AntC and x2 = ΔHv was created and this was fitted to determine the coefficients A and B shown in Equation (8) to give Equation (9):(9)$$\log S_{\mathrm{EN}}=\;-131\;\left(\pm 20\right)+0.5\;\left(\pm 0.1\right)\mathrm{AntC}+1.03\;\left(\pm 0.16\right)\;\Delta H_{V}$$

Table 1 (final column) shows the predicted values of sensitivity calculated from Equation (9) and these are plotted as a correlation between the experimental values obtained for sensitivity in Figure 6b. This indicates a good correlation between the experimental values and the fitted values with the largest deviation being for n-butylamine. The highest sensitivity is seen for dibutylamine and the lowest to triethylamine.

### 3.5. Effect of Water Vapour

For the practical applications of OFET chemical sensors they need to operate typically in environments where humidity levels may vary. Danesh et al. [27] previously showed that DPP-T-TT OFETs are able to sense ammonia over a wide range of relative humidity levels (RH) and that the OFET devices were stable in their responses under these conditions over several weeks. For practical applications, as described in the introduction, it is important to understand the dynamic range of the sensors as well as the influence of interferents. Figure 5a illustrates once such a concentration–response curve for up to 200 ppm ammonia in dry air. Here, this study was expanded to determine the sensitivity of the OFET sensors while measuring different concentrations of ammonia up to between 20 to 100 ppm in carrier air that ranged from 0 to 80% RH. As shown in Figure 9a, which illustrates the I_DS_ when the sensors were exposed to different % RH from a dry air baseline, the OFET response to water vapour alone is barely measurable up to 60% RH and is just above the noise level of the sensors. From the different concentration–response curves for ammonia, the sensitivity at each humidity level was extracted from the data. Data from one individual OFET are plotted in Figure 9b, indicating that the apparent sensitivity of the device to ammonia increased proportionately with relative humidity.

## 4. Discussion

OFET devices fabricated on flexible substrates using solution processing techniques with DPP-T-TT as the semiconductor and a high k dielectric layer were found to operate reliably at low V_GS_ and V_DS_ (−3 V), giving a threshold voltage of −0.5 V. To fabricate a viable OFET gas sensor requires an OSC that has high stability in air and is resistant to moisture. The DPP-based polymers have good characteristics and are relatively stable with little response to changes in humidity as shown in Figure 9a. Devices do display drift and decreasing I_DS_ over time, as shown in Figure 5c. However, the device lifetime is good for solution processed devices and it has been possible to make repeated measurements over several months using the same devices. Devices fabricated in the same batch may have different base I_DS_ due to variations in the fabrication process such as the thickness of the OSC and the dielectric layers. Despite this variation in the initial current between devices, the measurement of the % change in I_DS_ when responding to volatile analytes is robust and gives repeatable results. As these devices are made by solution processed techniques on flexible substrates, they are not intended to compete with established sensor technologies but to open new potential applications for potentially cheap sensors that can be deployed in large numbers for brief times or for scenarios where single use sensors are the norm, such as in medical applications.

A major contribution of this paper to the state of the art is the attempt to correlate the sensor responses with the molecular characteristics of the analytes tested. The molecular properties of the OSC that influence the strength of the interaction with the volatile analytes that result in the transduction of a signal include lipophilicity, polarizability, electron density and thin film morphology. For the analyte, the size, volatility and whether it is hydrophobic or hydrophilic will influence the adsorption and desorption in the OSC and dielectric layers. The lipophilic parameter of dibutylamine (cLogP = 2.83) is higher than that of n-butylamine (cLogP = 0.97) and this may influence the interaction with DPP-T-TT (cLogP = 22.695) and account for the higher sensitivity to this compound. A study by Bissell et al. [38] attempted to relate analyte volatility across a wide range of compounds to the responses of conducting polymers used as resistive gas sensors, on the basis of the non-specific partitioning of analyte vapours into the organic polymers. It was demonstrated that, for analyte functional classes (alcohols, esters, alkanes, and hydrocarbon-only aromatics), the electrical resistance changes of various polypyrrole and polythiophene-based conducting polymer sensors conform to a linear relationship with vapour concentration, producing a fixed amplitude of sensor response at an analyte saturated vapour pressure. It was shown that if C_v_ is the vapour concentration required to produce a fixed level of response in the sensor for a series of compounds that share the same functional group, e.g., alcohols, it can be shown that a relationship exists between C_v_ and the boiling point of the analyte T_b_. This is somewhat analogous to the findings presented here that relate the interactions of ammonia and alkyl-amines with the OSC used in the OFET device, where Antoine’s C constant (AntC) in degrees Celsius and the heat of vaporization at boiling point (ΔH_V_) can be correlated with the sensitivity of the OSC to the amines tested. Transduction resulting in a change of I_DS_ may occur through the doping/dedoping of the organic semiconductors, trapping/quenching of charge carriers, alteration of the molecular arrangement of the active layer, or influence on the charge injection/extraction/transport at the various interfaces—surface, bulk or OSC-dielectric interfaces. The current reduction observed (∆I_DS_) on exposure to ammonia to polymers such as polythiophene is generally attributed to lone pair of electrons of ammonia which influences the charge transport in the OSC, forming linkage type structures with the organic molecules, and trap/dedope at the polymer semiconductor/dielectric interface which negatively shifts the threshold voltage (∆V_T_) [13]. It is likely that similar mechanisms will also operate in the DPP-based OSC, but the substituent groups of amines will also define and influence the observed ∆I_DS_. Here, we demonstrated that these devices have exquisite sensitivity to ammonia and amines with an LOD at ppb levels. It is likely that the size, shape and charge influence the adsorption of the molecules to the OSC. While it is clear that while we do not yet have enough information to understand the exact nature of the analyte interactions that lead to the transduction of a signal in an OFET-based gas sensor, we now have an empirical method of predicting how they behave to different analytes based on the model presented here, based on the Antoine equation. Hence, in order to create new gas sensors tailored to the detection of analytes of interest, it is possible to devise a strategy of substitution of the 2-octyldodecyl groups labelled as R in Figure 1b with appropriate hydrophilic or hydrophobic groups, that add low concentrations dopant molecules to the polymer to modify the band gaps within the material, and of course to explore the wide variety of OSCs that are now available. In all cases it would be possible to set up a matrix of experiments to characterise the materials based on the QSAR principles described without having to explore each possible variant.

The enhanced response to ammonia when humidity levels are increased reflects the complexity of ammonia and its interaction with water. NH_3_ is understood to accept hydrogen bonds with water through its lone pair, forming a molecular complex H−O−H···NH_3_ with a comparatively strong hydrogen bond. Various studies [39,40] agree on a hydrogen-bonded structure with the ammonia molecule donating its lone pair of electrons and the water molecule accepting them with one of its OH groups. The hydrogen bond is almost linear, but the ammonia molecule is significantly tilted and the complex is dynamic. The water unit interchanges hydrogen atoms and the ammonia unit rotates almost freely. Hence, under humid conditions, the ammonia–water clusters are the active adsorbent onto the OSC and the size and complexity of these clusters may be dependent on the humidity level. The OSC is in effect interacting with different dynamic clusters of ammonia–water complexes at different RHs. Further study is required to understand these interactions and how they influence the charge carriers in the OSC.

## 5. Conclusions

The mechanism by which OFET sensors interact with volatile analytes is poorly understood and analytes such as ammonia and alkylamines can interact with OSCs through many possible pathways to influence the transduction process. This study demonstrates that despite the complexity of both the analytes and the transduction pathways of the OFET device, it is possible to model the response and sensitivity of the device to a range of amines through the Antoine constant C and the heat of vaporization at the boiling point. This simple model allows the prediction of how OFET devices may respond to volatile analytes and will allow the design and testing of OSC devices with more specific responses to selected chemical species. OFET devices based on DPP-T-TT have great potential applications for the detection of ppb levels of amines that surpass many other commercial sensors.

## Figures and Tables

**Figure 1 sensors-21-00013-f001:**
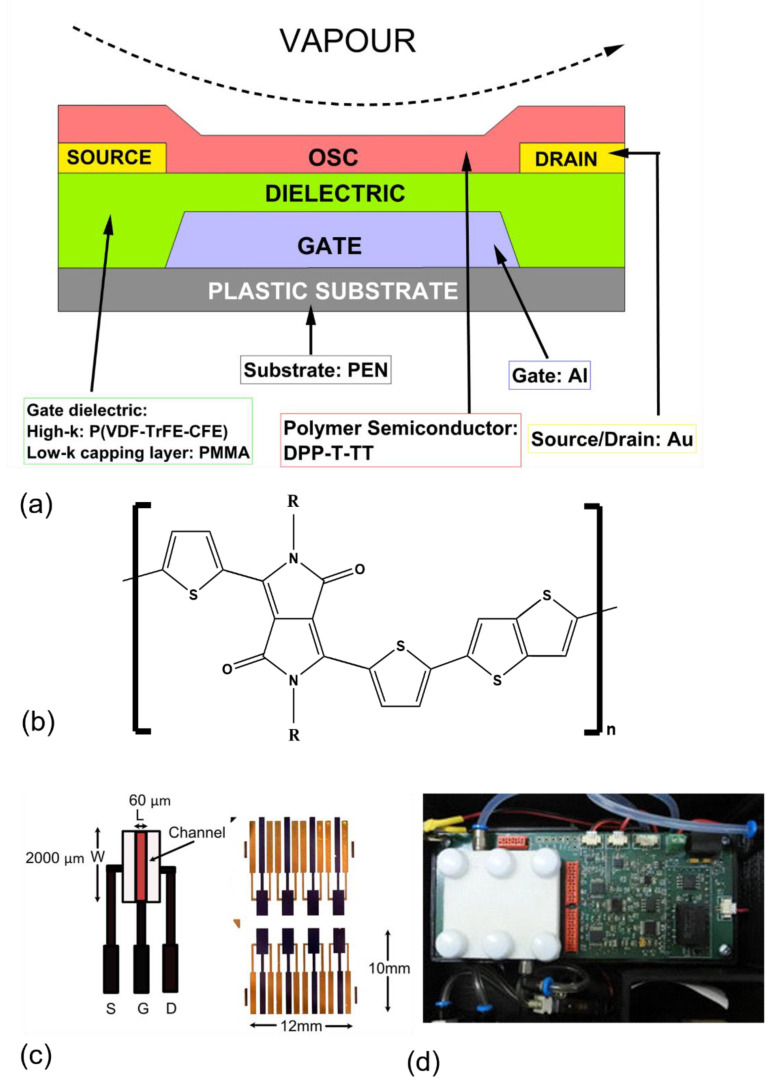
(**a**) Cross section of bottom-gate bottom-contact organic field effect transistors (OFETs) device (BGBC); (**b**) Chemical structure of organic semiconducting polymer (OSC) DPP-T-TT (R = 2-octyldodecyl); (**c**) (**left**) single OFET with connection pads and showing the channel (L = length, W = width), (**right**) array of 8 OFETS fabricated on the polyethylene naphthalate (PEN) substrate (**d**) data acquisition system—the white header covers the OFET array and serves as a gas delivery system.

**Figure 2 sensors-21-00013-f002:**
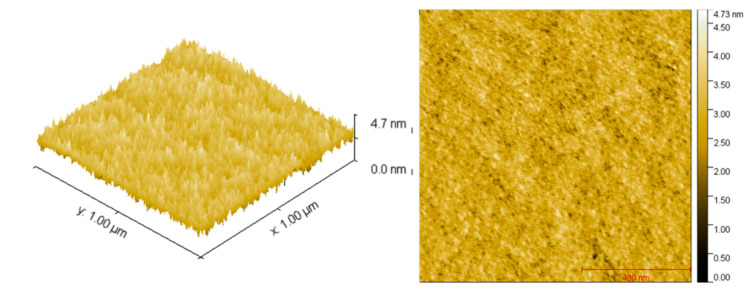
Atomic force microscopy images of the DPP-T-TT film deposited by spin coating and measured using tapping mode.

**Figure 3 sensors-21-00013-f003:**
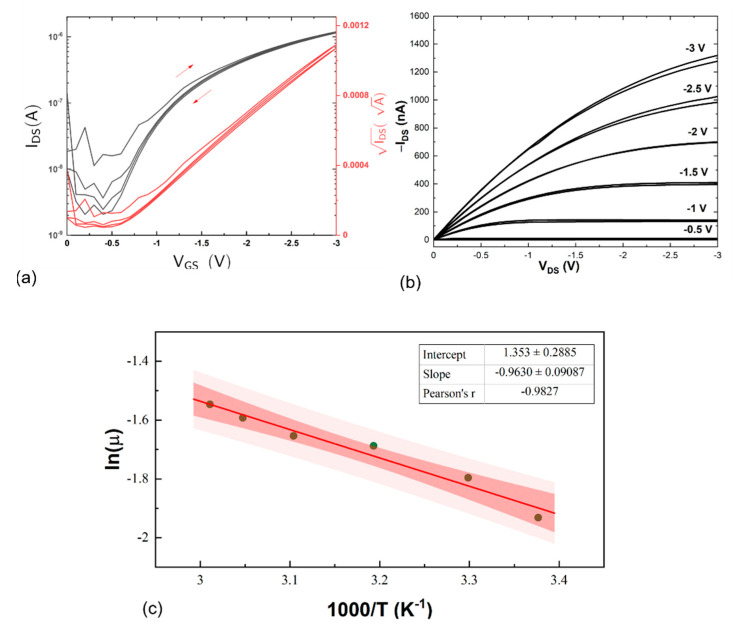
(**a**) Transfer curve used to establish the operating characteristics of the OFET devices—the devices were scanned 4 times between 0 and −3 V V_GS_ to observe the hysteresis of the devices; (**b**) the output curve at different V_GS_ voltages between 0 to −3 V; and (**c**) the Arrhenius plot of the dependence of mobility with temperature established an E_a_ of 82 mV.

**Figure 4 sensors-21-00013-f004:**
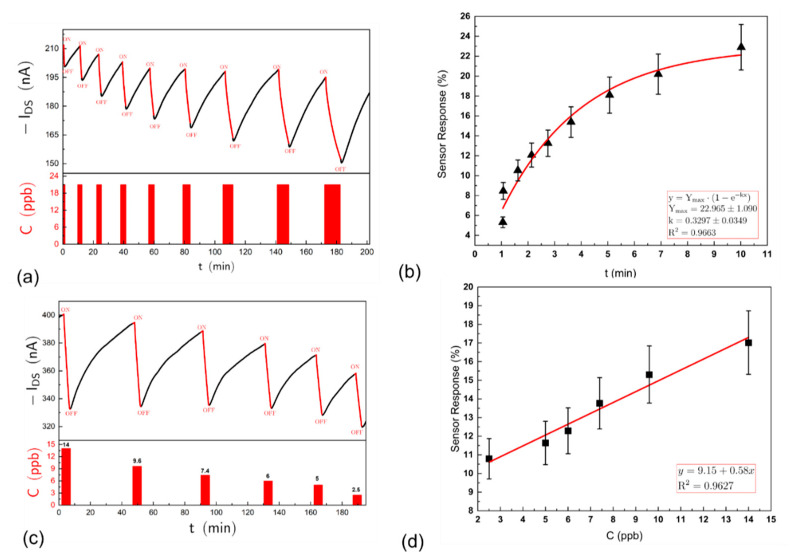
(**a**) Exposure of an OFET sensor to 21 ppb ammonia for increasing lengths of time, followed by recovery in clean air. Red bars in the lower panel indicate the period of exposure. The sensor response increases with the length of exposure; (**b**) the kinetics of the response to a fixed concentration of ammonia over time. The response approaches a steady state after about 10 min but has not yet reached an asymptope; (**c**) the top panel—another OFET sensor was exposed to pulses of ammonia at different concentration steps for periods of 5 min followed by recovery in clean air for 30 min. The change in I_DS_ current was measured at each step. Bottom panel—concentration steps in ppb presented to the sensor; (**d**) concentration–response curve for ammonia at low concentrations from repeated experiments shown in (**c**)—this can be fitted by a linear equation.

**Figure 5 sensors-21-00013-f005:**
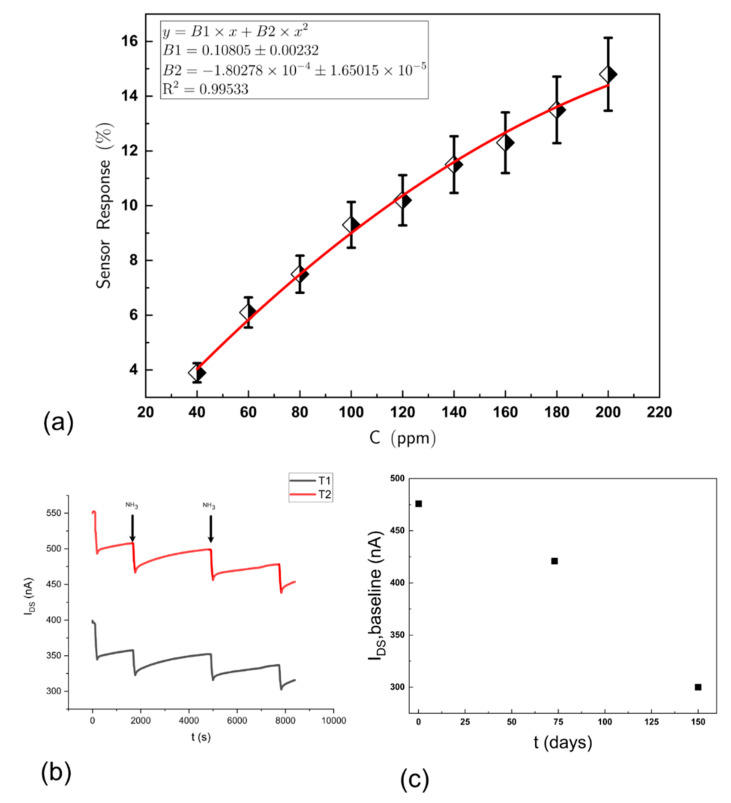
(**a**) Concentration–response curve for ammonia between 40 and 200 ppm. This is best fitted by a second order polynomial equation; (**b**) raw data responses from two OFET devices to repeated exposure to ammonia; (**c**) baseline I_DS_ for one device monitored over 5 months showing a decrease in current but a still usable device after this period.

**Figure 6 sensors-21-00013-f006:**
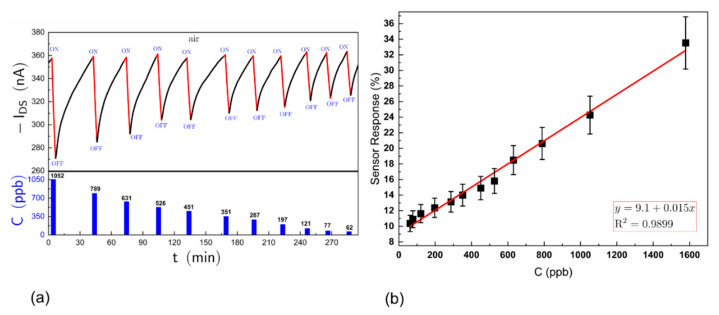
(**a**) Top—raw responses to a series of pulses of triethylamine vapour in dry air. Bottom—duration of vapour pulse sequence between 1052 and 62 ppb; and (**b**) concentration–response curve for triethylamine vapour.

**Figure 7 sensors-21-00013-f007:**
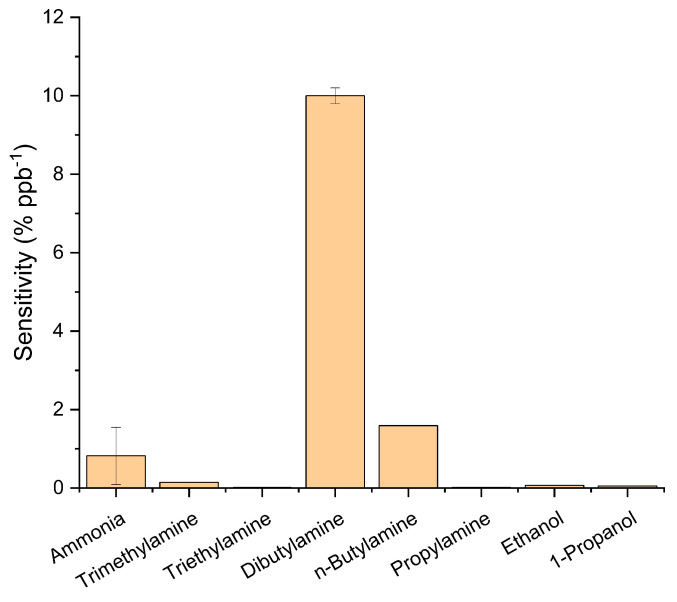
Plot of measured sensitivities of DPP-T-TT OSCs to different analytes, indicating high selectivity to the dibutylamine over the other analytes tested.

**Figure 8 sensors-21-00013-f008:**
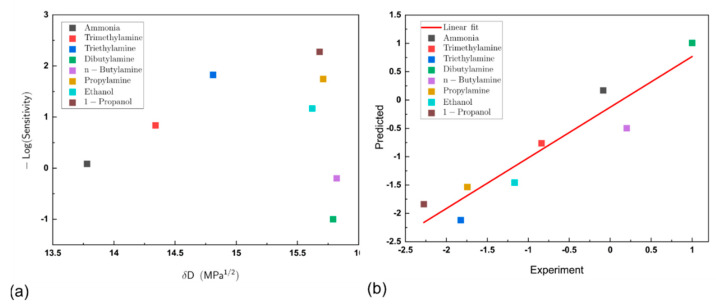
(**a**) Quantitative structure–activity model (QSAR) plot of log (sensitivity) versus the dispersion force (δD) for each analyte tested. No coherent trend can be seen; (**b**) good correlation is seen between the experimentally derived sensitivities for a series of amines with the predicted values from the model based on the Antoine constant C and the heat of vaporization at boiling point (Equation (9)).

**Figure 9 sensors-21-00013-f009:**
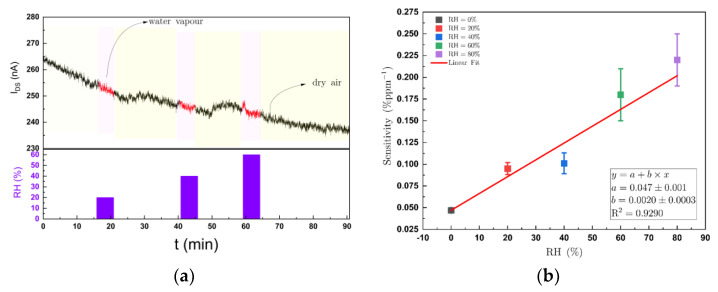
(**a**) Plot of I_DS_ for an OFET device when exposed to different humidity levels. The baseline was in dry air, the lower panel illustrates when different changes in humidity were applied and the red portion of the I_DS_ trace above highlights the current during this exposure; (**b**) the apparent increase in OFET sensitivity to ammonia with increasing relative humidity (RH). A series of concentrations of ammonia ranging from 20 to 100 ppm were presented to the OFET sensor while varying the RH from 0 to 80%. The I_DS_ was measured at V_GS_ of −3 V and V_DS_ of −3 V. From the resulting concentration–response curves at each humidity step, the gradients of the curves were extracted and plotted here as sensitivity versus %RH.

**Table 1 sensors-21-00013-t001:** Molecular parameters, Antoine parameters, measured sensitivity, limit of detection (LOD) and predicted sensitivity.

Compound	δD	δP	δH	AntC °C	ΔH_V_ J.M^−1^	Measured Sensitivity %ppb^−1^	Measured LOD ppb	Predicted Sensitivity %ppb^−1^
Ammonia	13.78	16.74	18.82	235.9	23.37	0.82 ± 0.73	2.17	0.99
Trimethylamine	14.34	2.86	4.39	233.8	23.39	0.145 ± 0.008	2.9	0.14
Triethylamine	14.81	2.77	2.9	216.2	29.83	0.015 ± 0.001	34.5	0.013
Dibutylamine	15.79	2.68	4.4	200.4	39.83	10.00 ± 0.20	0.025	14.73
n-Butylamine	15.82	4.6	8.38	215.1	31.89	1.59 ± 0.01	0.056	0.41
Propylamine	15.71	5.15	8.33	219.1	29.12	0.018 ± 5 × 10^−4^	129	0.041
Ethanol	15.62	9.3	17.19	202.8	36.38	0.068 ± 0.007	116	0.075
1-Propanol	15.68	7.34	14.59	197.4	38.39	0.053 ± 2.4 × 10^−4^	226	0.017

## Data Availability

The data presented in this study are available on request from the corresponding author.

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
