# Peer review of "Amine Detection Using Organic Field Effect Transistor Gas Sensors"

_sensors, 2020, doi:10.3390/s21010013_

Round 1

Reviewer 1 Report

This paper needs to be improved in the following aspects:

(1) What are the main contributions of this paper and the current research situation need to be described in detail to help readers understand the value of the author's research.

(2) To what extent does the current sensitivity of amine gas detection reach and how much does this study improve the current sensitivity?

(3) The sensor based on other principles of amine Gas detection can provide the same accuracy as the Gas Sensors in the Organic Field Effect Transistor. What is the purpose of this study?

(4) Figure 1 (b) and Figure 7 are not very clear. Please improve the resolution.

Author Response

(1) What are the main contributions of this paper and the current research situation need to be described in detail to help readers understand the value of the author's research. 

Addressed in revised introduction. Please see changes in red.

(2) To what extent does the current sensitivity of amine gas detection reach and how much does this study improve the current sensitivity?

Addressed in revised introduction and discussion. Please see changes in red.

(3) The sensor based on other principles of amine Gas detection can provide the same accuracy as the Gas Sensors in the Organic Field Effect Transistor. What is the purpose of this study?

We show much higher sensitivity over commercially available ammonia gas sensors based on other technologies.  This is clarified in the introduction and discussion. The purpose of this study was a systematic study of OSC interactions with ammonia and amines resulting in an empirical model of predicting how the sensors will react to a particular analyte.

(4) Figure 1 (b) and Figure 7 are not very clear. Please improve the resolution.

Figures redrawn

Reviewer 2 Report

The manuscript entitled “Amine Detection using Organic Field Effect Transistor Gas Sensors” describes a gas sensor based on organic field-effect transistors tailored for ammonia and a series of allylamines. One of the major flaws of this manuscript is the poor discussion about the selectivity of the proposed device. In fact, the authors described (lines 346 and 347) and discussed (lines 401-412) a possible interference of relative humidity (RH), but they do not discuss how to deal with this in practical situations. Furthermore, the influence of other potential species was not even discussed. In my opinion, the development of a gas sensor must be validated against real samples. Despite being interesting, the authors should test the sensor with further experiments. Therefore, the manuscript should be rejected in the present form.  Some comments:  - The abstract could contain the results obtained, such as LOD.  - What is the novelty of this work regarding analytical performance? A table should be provided with a comparison among similar gas sensors -  Figure 4d. Error bars must be provided. Why the r2 is so poor? 

Author Response

1. The manuscript entitled “Amine Detection using Organic Field Effect Transistor Gas Sensors” describes a gas sensor based on organic field-effect transistors tailored for ammonia and a series of allylamines. One of the major flaws of this manuscript is the poor discussion about the selectivity of the proposed device.

Additional analytes inserted in Table 1 which defines the selectivity of the devices to ammonia, amines and two alcohols. A new Figure 7 has been added to demonstrate the relative selectivity and apart from ammonia and amines, two alcohols have been added to the data set.

2.  In fact, the authors described (lines 346 and 347) and discussed (lines 401-412) a possible interference of relative humidity (RH), but they do not discuss how to deal with this in practical situations.

New figure 9(a) added to demonstrate that there is negligible response to humidity by the semiconductor.

3. Furthermore, the influence of other potential species was not even discussed. In my opinion, the development of a gas sensor must be validated against real samples.

Additional analytes added to data presented in Table 1.

4. Despite being interesting, the authors should test the sensor with further experiments. Therefore, the manuscript should be rejected in the present form.  Some comments:  - The abstract could contain the results obtained, such as LOD.  - What is the novelty of this work regarding analytical performance? A table should be provided with a comparison among similar gas sensors

Analytical performance and comparison with other sensors were not the aim of this paper which focusses on a model to predict how the semiconducting material will respond to different types of analytes – we indicate the measurable ammonia range by commercial sensors in a modified introduction.

5.  Figure 4d. Error bars must be provided. Why the r2 is so poor? 

The Figure has been replotted with error bars.

We would argue that R2 is not poor considering the extremely low concentration of analyte that is being presented to the sensor.

Reviewer 3 Report

The authors examine the response of OFETs based on DPP-T-TT to ammonia, both in the presence and absence of humidity. The responses to a series of alkyl-amines, trimethylamine, triethylamine, n-propylamine, n-butylamine and dibutylamine over a range of concentrations were investigated. It is observed that the highest sensitivity is to dibutylamine and the lowest is to triethylamine. The similarity between the predicted sensitivity and the measured results is impressive. This result is publication worthy.

There are nonetheless several points that need to be reconsidered in the manuscript before it is publication ready.

Most importantly, clarification is needed how exactly the authors attained equation 8. As these results make up a substantial part of the manuscript more detail should be given to how the calculations were done and from which sources the values were taken.

Different configurations of OFETS are possible, the authors could specify that they are using a bottom-gate-bottom-contact configuration. Is there a particular reason why the authors chose this configuration? 

In the introduction the authors thoroughly explain the need for gas sensors in general and the advantages of OFETs versus more convention metal oxide based sensors. The authors also present the importance of ammonia as a test gas and why other organic based systems need optimization. What is missing from the introduction, however, is the rationale for why DPP-T-TT was selected for this study.

Figure 1c shows a picture of 8 OFETS fabricated on a PEN substrate, and d shows the gas delivery system. In both cases, it would be interesting to have a scale for the pictures.

The authors report that  The response to ammonia results in a r decrease in IDS and that when exposed to a fixed concentration of ammonia (21 ppb) for varying periods of time followed the change in IDS is proportional to the length of exposure. The authors plotted the sensor response to ammonia over a ten minute period Fig, 4 b. It would be more appropriate to explain how the response was calculated in the experimental section than in the figure caption. In this case did the authors take the Ids values during the measurements done under different exposure times or were only the results attained during the final 10 minute exposure? In either case how would the values compare, is there significant variation?

In Figure 4 c and d the authors show the response to different concentrations of ammonia after 5 minute exposure. The authors find a linear relationship between 2.5 and 14 ppb ammonia. If the  authors add the 21 ppb from the previous measurements has the sensor then reached saturation?

Overall the sensor appears to not fully recover after exposure to Ammonia. Did the authors ever do repeated exposures under the same conditions to see if the reproducibility of the sensor signal?

In the discussion section the authors state that the device lifetime is good for solution processed devices and that it was possible to make repeated measurements over several months using the same devices. An additional graphic with these results would be very significant.

The authors properly state that in most real applications the sensors need to operate steadily despite changes in background humidity For these measurements the authors chose much higher concentrations (although more fitting for the applications mentioned in the introduction) than the low ppb ranges examined before. What is the rationale for this variation? The authors should add an explanation.

Some small things, bracket missing around % in Figure 4 b, line 255 typo the s missing in response. Equation is sometimes written out or abbreviated this should be consistent throughout the manuscript.

Author Response

1. Most importantly, clarification is needed how exactly the authors attained equation 8. As these results make up a substantial part of the manuscript more detail should be given to how the calculations were done and from which sources the values were taken.

We had given a reference 31 to the source of data but perhaps this was not clear – a few words have been added to clarify this.

2. Different configurations of OFETS are possible, the authors could specify that they are using a bottom-gate-bottom-contact configuration. Is there a particular reason why the authors chose this configuration? 

The rationale for choosing a bottom-gate-bottom-contact configuration is explained in the introduction.

3. In the introduction the authors thoroughly explain the need for gas sensors in general and the advantages of OFETs versus more convention metal oxide based sensors. The authors also present the importance of ammonia as a test gas and why other organic based systems need optimization. What is missing from the introduction, however, is the rationale for why DPP-T-TT was selected for this study.

The rationale for choice of DPP-T-TT as a semiconductor on the basis of air stability and previous work done on ammonia sensing is explained in the introduction.

4. Figure 1c shows a picture of 8 OFETS fabricated on a PEN substrate, and d shows the gas delivery system. In both cases, it would be interesting to have a scale for the pictures.

A scale has been added to the photograph of 8 OFETS.

5. The authors report that  The response to ammonia results in a r decrease in IDS and that when exposed to a fixed concentration of ammonia (21 ppb) for varying periods of time followed the change in IDS is proportional to the length of exposure. The authors plotted the sensor response to ammonia over a ten minute period Fig, 4 b. It would be more appropriate to explain how the response was calculated in the experimental section than in the figure caption.

The explanation of the response calculation has been added to the experimental section.

6. In this case did the authors take the Ids values during the measurements done under different exposure times or were only the results attained during the final 10 minute exposure? In either case how would the values compare, is there significant variation?

For ammonia and amines, the change in IDS is proportional to the length of exposure to the analyte. We decided to fix the exposure to 5 minutes for all of the experiments described and this statement has been added to clarify the ambiguity in line 290. 

7. In Figure 4 c and d the authors show the response to different concentrations of ammonia after 5 minute exposure. The authors find a linear relationship between 2.5 and 14 ppb ammonia. If the  authors add the 21 ppb from the previous measurements has the sensor then reached saturation?

No the sensor did not reach saturation ... an additional Figure 5 has been added to show concentration-response up to 200 ppm.

8. Overall the sensor appears to not fully recover after exposure to Ammonia. Did the authors ever do repeated exposures under the same conditions to see if the reproducibility of the sensor signal?

Additional Figure 5 b added to demonstrate repeated exposures 

9. In the discussion section the authors state that the device lifetime is good for solution processed devices and that it was possible to make repeated measurements over several months using the same devices. An additional graphic with these results would be very significant.

Figure 5 c added to show the decrease in baseline IDS over 5 months and this is discussed in lines 299-311

10. The authors properly state that in most real applications the sensors need to operate steadily despite changes in background humidity. For these measurements the authors chose much higher concentrations (although more fitting for the applications mentioned in the introduction) than the low ppb ranges examined before. What is the rationale for this variation? The authors should add an explanation.

Added Figure 9 a to demonstrate little response to changes in humidity.

Lines 402-405 added for rationale.

11. Some small things, bracket missing around % in Figure 4 b, line 255 typo the s missing in response. Equation is sometimes written out or abbreviated this should be consistent throughout the manuscript.

Typographical errors were corrected.

Reviewer 4 Report

In this paper, a model for the gas sensitivity of organic semiconductor thin film transistors is introduced. While the mechanisms of the gas sensing response in OSC TFTs are rather complex, the model is very simple, yet accurate enough according to the results shown. I consider that this paper should be accepted for publication in Sensors after the following minor aspects are addressed:

  1. Could your model generalise well also to oxidising species?
  2. Your experimental procedure enables you to obtain an array of OSC TFTs. Since all TFTs in this paper have the same gas sensitive polymer coating, could you add some results on device-to-device reproducibility?
  3. Could you also add some results on the long term stability of your devices?
  4. Finally a short discussion (one or two sentences) about how to ameliorate the selectivity experienced would be of interest.
  5. Make sure you always leave a blank space between a number and its units.

Author Response

1. Could your model generalise well also to oxidising species?

We have expanded Table 1 to include two alcohols showing that the model is equally valid. Initial lab experiments indicate negligible response to oxidising gases such as NO2. This is not part of this paper.

2. Your experimental procedure enables you to obtain an array of OSC TFTs. Since all TFTs in this paper have the same gas sensitive polymer coating, could you add some results on device-to-device reproducibility?

New figure 5 (b) added showing raw response data from 2 separate devices.

3. Could you also add some results on the long term stability of your devices?

New figure 5 (c) added showing change in IDS over 5 months.

4. Finally a short discussion (one or two sentences) about how to ameliorate the selectivity experienced would be of interest.

Done - lines 475-482

5. Make sure you always leave a blank space between a number and its units.

Done

Round 2

Reviewer 2 Report

The authors have provided suitable replies for all my concerns. The paper can be now accepted for publication.